# Electrospun Fibrous Membrane with Confined Chain Configuration: Dynamic Relaxation and Glass Transition

**DOI:** 10.3390/polym14050939

**Published:** 2022-02-26

**Authors:** Nuozi Zhang, Chenhong Wang, Hao Chen, Jiaen Wu, Charles C. Han, Shanshan Xu

**Affiliations:** 1Institute for Advanced Study, Shenzhen University, Shenzhen 518060, China; zhangnuozi@sina.com (N.Z.); wangchenhong@iccas.ac.cn (C.W.); mason.hao.chan@gmail.com (H.C.); jiaen_wu1101@foxmail.com (J.W.); c.c.han@iccas.ac.cn (C.C.H.); 2Key Laboratory of Optoelectronic Devices and Systems of Ministry of Education and Guangdong Province, College of Physics and Optoelectronic Engineering, Shenzhen University, Shenzhen 518060, China; 3State Key Laboratory of Polymer Physics and Chemistry, Joint Laboratory of Polymer Science and Materials, Beijing National Laboratory for Molecular Sciences, Institute of Chemistry, Chinese Academy of Sciences, Beijing 100190, China

**Keywords:** physical aging, relaxation, electrospun membrane, PLGA, MCT

## Abstract

Thermodynamic glass transition processes of electrospun membranes were first introduced to study their dynamic relaxation nature, which is not constantly in equilibrium. The relaxation modes of electrospun membranes are slow but measurable near and above the T_g,_ given the stretched chain over long distances. Based on differential scanning calorimetry (DSC) experiments and the general principle of mode-coupling theory (MCT), endothermic peak temperature and relaxation enthalpy were used to analyze the relaxation process by capturing these instantaneous “arrested” structures. The short- and long-wavelength relaxation modes could be identified with different annealing times and temperatures relative to DSC-measured T_g_ for electrospun membranes with different molecular weights. Results clearly showed the dynamic nature of a glass transition in polymeric materials. T_p_ and enthalpy loss initially increased and then directly decreased with the increase in annealing time. When T_a_ > T_g_, regardless of the size of the molecular weight, the T_p_ and enthalpy loss of the PLGA fibers would directly decrease, and the curves would shift toward the melted one. Combination of electrospinningand normal DSC instrument can be used to investigating the dynamic relax process through an adequately designed kinetic scanning procedure. This result can be explained by the general principle of MCT-type dynamic theory.

## 1. Introduction

Physical aging of glassy polymer materials, which is below their transition temperature [1] during storage and application, leads to changes in the electrical, thermal and mechanical properties [2,3,4]. Shifts in material properties lead to further changes in material functionality, especially those designed for long service life [1,2,3,4,5]. A popular example is polyurethane, which is used to insulate pacemakers [6,7]. Aging and degradation of the polyurethane coating lead to insulation failure and leakage in pacemakers, which can cause muscle irritation [8]. Another example is poly(lactide-glycolide) (PLGA), a biodegradable polymer used to make absorbable sutures for wound healing [9,10,11,12]. Physical aging and degradation are critical for application performance and storage processes of such polymers [13,14].It is also crucial for shape memory polymers, which are intelligent materials, rely on the different stress relaxation models. Polymers can be easily deformed into a temporary shape in one state and then quenched into a designed permanent shape by receiving heat or other stimuli [15,16,17].

Most simulations experiments based on accelerated aging are unreliable because accelerated aging does not always follow a linear equation with time and temperature as variables. Aging experiments typically require extended periods of time at temperatures far below the T_g_ measured by differential scanning calorimetry (DSC). Otherwise, the amount of the endothermic change or time scale during this change would be difficult to be observed for analysis, given the sensitivity of the DSC instrument [18]. Mode-coupling theory (MCT) [19], is one of the most popular theories to study glass transition of structural materials [3,20,21,22]. N relaxation times constantly appear for an N-particle system in MCT. Götze MCT [23] has truncated all high-order correlation terms after the second term (fourth-order correlation in the reciprocal space) because of a complicated operation. Although Götze MCT [24] is a truncated theory, it only provides two modes (α and β) for glass transformation and demonstrates an accelerated slowdown of the α branch (or mode) relative to the β relaxation time in a simple system. However, the importance of the dynamic modes and the slowdown of long wavelength modes [25] are the most important characteristics of any glassy materials in a temperature below the DSC-measured T_g_ and in the release of the shape memory function in applications for a glassy material. Although any MCT theory for understanding the glass transition process includes Götze MCT [24] failed to provide detailed quantitative guidance for data analysis, multiple relaxation times are essential. according to the general principle of MCT, extensive relaxation times slow down non proportionally at temperatures lower than the DSC T_g_. This DSC T_g_ is a convenient temperature point, which marks the emergence of an observable endothermic change rate of the testing specimen at the DSC scanning rate.

This work selected PLGA-regenerated film and electrospun membrane to study dynamics relaxation during the glass transition near the DSC-defined T_g_. The packing density of the regenerated film is higher than the electrospun membrane, but the electrospun membrane has a large surface-to-volume ratio base on a uniform nanofiber morphology. This electrospun membrane’s relaxation process (spectrum or modes) was vastly extended as PLGA chains are stretched, constrained, and confined in these fibers. The polymer chains are physically constrained and confined in the electrospun fibers of the membrane to a certain scale. Thus, the polymer chains are not only important in a membrane application but also a favorable model for studying the glass transformation and aging processes in general. Moreover, PLGA electrospun membranes with different molecular weights are studied to investigate the relaxation process in various timescales given their different chain mobilities, which are coupled into the long-distance relaxation modes. The relaxation process is influenced by the local structural environment, which varies at different annealing times and temperatures. The process can be accelerated by increasing the annealing temperature, thereby obtaining the DSC curves and reflecting the chain length and segmental mobility. Therefore, through this study, instantaneous structures during the DSC measurements over different annealing times and temperatures can be obtained to understand better the dynamic nature of glass transition and a favorable control of its structure/morphology during manufacturing, storage, and shape memory function delivery in the application.

## 2. Experimental Section

### 2.1. Materials

PLGA (Mw = 20,000 g/mol [20 k], 60,000 g/mol [60 k], 100,000 g/mol [100 k]; LA/GA = 75/25, mol/mol) was purchased from Jinan Daigang Biology Engineer Co., Ltd., Jinan, China. N, N-dimethyl formamide (DMF), acetone, chloroform, and deuterated chloroform were obtained from the China National Medicines Corporation, Ltd., Beijing, China. All other reagents used were of commercial analytical grade.

### 2.2. Fabrication of PLGA-Regenerated Film and Fibrous Membrane

PLGA solutions were prepared in a mixed solvent system of acetone and DMF (*v*/*v* = 5/5). The 20 k, 60 k, 100 k membrane electrospinning process was performed in a sterile environment at 16, 20, 24 kV and a steady flow rate of 20, 15, 10 μL/min (spinneret with an inner diameter of 0.3 mm). The electrospun fibers were collected on a metal drum (as an electrode; diameter of 9 cm; tip-to-collector distance of 18, 23, 28 cm), rotating at approximately 120 rpm. The thickness of the fibrous scaffolds was set to 100 ± 10 µm by controlling the spinning time. The “regenerated film” was prepared by melting the electrospun membrane (a PLGA fibrous membrane with an Mw of 60 k was placed between two flat templates and was heated up to 65 °C for 6 h) and was used for further studies for easy comparison with a fibrous membrane in aging and other experiments. The DSC measurement was conducted to ensure that the treatment had completely melted the fibrous structure and provided a regenerated film. To remove the residual solvents, all the samples were further dried under 8–10 °C in the vacuum for 1 month to remove the residual solvents.

### 2.3. Characterization

All characterizations were performed after removing the residual solvents.

The morphologies of the electrospun fiber were observed using a scanning electron microscope (JEOL JSM-6700F, Tokyo, Japan) at an accelerating voltage of 5 kV. Samples were sputter-coated with platinum before analysis.

The water contact angles of the PLGA electrospun membrane were measured with a sessile drop method using a digital contact angle measurement system with a CCD camera (Powereach JC2000A, Shanghai, China). A water droplet of 5 µL was used and a snapshot of the image was taken to measure the static contact angle.

Molecular weight and molecular weight dispersity were measured through gel permeation chromatography (GPC). The measurements were performed with a chloroform system equipped with a differential refractometer as a detector. Chloroform was fluent at 35 °C with a 1 mL/min flow rate. Molecular weights were calibrated with PS standards.

The ^1^H-NMR spectra of components were recorded using a Bruker Avance 400 spectrometer operated at 400 MHz, with deuterated chloroform (CDCl_3_) as the solvent. ^1^H-NMR (400 MHz, CDCl_3_, 25 °C): δ 5.31–5.11 (m, 1H; C*H*(CH_3_)C(O)O), 4.92–4.57 (m, 2H; C*H*_2_C(O)O), 1.65–1.52 (m, 3H; CH(C*H*_3_)C(O)O).

Thermal properties, such as glass transition temperature and enthalpy loss of sample heated after physical aging or annealing, were determined through differential scanning calorimetry using a DSC (TA Instruments Q-2000, New Castle, DE, USA) apparatus, calibrated with indium in a nitrogen flow. The heating rate was 20 °C/min within −60–100 °C. Data were processed using the TA Universal Analysis v2.3 software(TA Instruments, New Castle, USA) and then reported after the first and second heating (re-heating) cycles.

The isothermal relaxation experiment was conducted following a specific procedure, as illustrated in Figure 1. The sample, which was weighed and sealed in aluminum pans, was annealed at a specific temperature (annealing temperature, T_a_ from 30 °C to 52.5 °C) for a specific time (less than 6 h) directly in the DSC pans. For the samples aged longer than 6 h (or in the drying and storage before measurements), the aging processes were performed outside an incubator. For measurements, samples were removed from storage and then quickly sealed and transferred to the DSC oven for further annealing and measurements following Figure 1. PLGA is a random copolymer. Thus, samples were checked through WAXS measurements at different states to ensure they all remained amorphous in all states.

## 3. Results and Discussion

PLGA was a US Food and Drug Administration-approved polymer that has been used as biodegradable materials for decades [9]. A PLGA electrospun fibrous membrane normally has a fiber diameter in the micrometer scale and a high surface area/volume ratio with enhanced properties. PLGA electrospun membranes with different molecular weights were studied to investigate the relaxation process in various timescales. The polymer chains of the electrospun fibers in the membrane were physically constrained and confined to a certain scale. The relaxation process was affected by the local structure and environment, such as different annealing times and temperatures. To generate DSC curves reflecting chain lengths and segment mobilities, the relaxation process can be accelerated by increasing the annealing temperature.

### 3.1. Preparation and Characterization of PLGA Electrospun Membranes

The compositions and properties of three different molecular weight PLGA materials were shown in Table 1 and Figure 1. These three PLGA materials have the same LA: GA ratio, confirmed by the NMR results in Figure 1b. The molecular weight distribution of PLGA in the fiber membrane was shown in Figure 1b, determined by GPC. Figure 1 also showed the morphologies and diameter distributions of PLGA electrospun membranes with different molecular weights. In Figure 1c, it can be seen that the surface of the PLGA electrospun membranes was smooth and the fiber diameters of PLGA electrospun with different molecular weights were uniform. With the increase of molecular weight, the average fiber diameter of the PLGA electrospinning fiber membrane increased as the viscosity of the electrospinning solution increased due to the risen of molecular weight (fiber diameter: 20 k = 1.20 ± 0.12 μm, 60 k = 1.39 ± 0.11 μm; 100 k = 1.39 ± 0.09 μm). Slight differences in fiber diameter hardly affect the experimental results. The hydrophilicity of the fiber membranes was evaluated by measuring the water contact angle of the electrospun membranes. The topology of the electrospun fiber membrane mainly contributes to the hydro-phobic properties of the material. The water contact angles of the three PLGA electrospun membranes with different molecular weights were all higher than 120° (water contact angles: 20 k = 128.68 ± 1°, 60 k = 126.14 ± 1°; 100 k = 123.96 ± 1°), which showed no significant difference in hydrophilicity.

### 3.2. Sample Pretreatment and Experimental Design

This study introduced PLGA-regenerated film and electrospun membranes to evaluate the relaxation dynamics below or around T_g_ during glass transition. All these PLGA samples were produced and dried at 8–10 °C in a vacuum for 1 month to achieve a comparable thermal history and limit the influence of residual solvent. All the samples were stored at 8 °C, far below the DSC T_g_ for 1 month, similar to the process exhibited in Figure 2a in the metastable region approaching the aging line (position below the “a” point). Traditionally, the dynamics of the glass transition of polymer material was a process from a thermodynamical equilibrium liquid state to a frozen-in thermodynamically non-equilibrium glass state [5,26,27,28]. When the liquid was cooled through the glass transition region, molecular mobility slows down, and the material may become a slow-varying but unstable glass (as represented by the solid red line in Figure 2a). The material system of this unstable state will tend to reach its equilibrium state (as reflected by the black dashed line in Figure 2a) through the relaxation process even after the temperature stops decreasing. At the reverse transition, glass transition was supposedly a thermodynamic equilibrium evolution, whereas all the relaxation processes aimed to follow the new temperature’s thermodynamic equilibrium. Depending on whether these modes were faster or slower than the changing temperature, the system can lag (as demonstrated by the purple and blue dashed lines in Figure 2a). In particular, the relaxation enthalpy measured from the relaxation of a stretched long chain was probably due to the energy released from segmental rotational and locally constrained configuration states. The relaxation entropy for the overall particle distribution cannot be determined directly by DSC measurements, either positively or negatively depending on the fluctuations of the intermolecular distances. Therefore, even with the same overall potential energy as a starting point (point “a” in Figure 2a on the purple or blue dashed line), the system may result in a different thermodynamic unstable state depending on the time-temperature path it has undergone. All samples were then loaded into the DSC instrument, and the measurements were performed, following the normal physical aging process through the DSC at different times and the blue dashed line displayed in Figure 2a.

The heating curves of the regenerated film and electrospun membranes were presented in Figure 2c. At the peak of the heating curve (T_p_), the endothermic temperature shifted to a higher temperature than the melt (representing the sample, which was heated to 373 K and quenched to 273 K to remove its thermal history then measured immediately). This phenomenon was mainly caused by solvent evaporation or densification, which has caused an increase in segmental packing density or a reduction in averaged intermolecular distance. In particular, this phenomenon may be considered the increase in potential energy or decrease in free volume [28,29]. Therefore, relaxation enthalpy was increased by increasing the annealing time, as depicted in Figure 2c. The initial inflection point of each curve (approximately 40 °C) represented the endothermic response that could be obtained using the DSC sensor. The starting shapes of each curve (from 40 °C to 45 °C) were relatively similar, indicating that the short wavelength modes were similar in regenerated film and fibers, and all these modes could catch up with the increase in temperature. The subsequent plots of fibers at higher than 45 °C resulted in a delayed thermal response, demonstrating a considerable shift to high temperatures. This result might be due to the long-wavelength modes constrained by the aligned chain entanglements in the fibrous sample, which cannot catch up with the DSC heating rate as quickly as the relaxed chain in the regenerated film during the reverse process.

Moreover, the fiber’s aligned polymer chain resulted in higher enthalpy relaxation than the regenerated film. The growth rates of the endothermic enthalpy were also similar for the regenerated film and fibers in the 40–45 °C range because the local or short-wavelength modes were similar. The growth rate of the T_p_ and relaxation enthalpy was gradually saturated with the increase in annealing time, as displayed in Figure 2d. When short and intermediate modes relaxed to their new equilibrium states, additional segmental pairs would reach their inter-molecular distance equilibrium distribution. Further annealing could only change the T_p_ and relaxation enthalpy from the long-wavelength modes, mainly due to the relaxation of stretched polymer chains.

### 3.3. The Effect of Annealing at a High-Temperature Region

The growth rates of the endothermic enthalpy were also similar for the regenerated film and fibers in the 40–45 °C range because the local or short-wavelength modes were similar. The growth rate of the T_p_ and relaxation enthalpy was gradually saturated with the increase in annealing time, as displayed in Figure 2d. When short and intermediate modes relaxed to their new equilibrium states, additional segmental pairs would reach their inter-molecular distance equilibrium distribution. Further annealing could only change the T_p_ and relaxation enthalpy from the long-wavelength modes, mainly due to the relaxation of stretched polymer chains.

After all the samples were stored at 8 °C for 1 month, experiments were conducted through DSC, following Figure 1. The DSC re-heating curve’s temporal evolution of PLGA samples after annealing at a certain T for a definite time was collected. The enhancement of the influence of the fibrous structure and extended relaxation modes on the electrospun membranes persists only for temperatures slightly below or above the T_g_ in Region II of Figure 2b. Figure 3a depicts the endothermic peak and relaxation enthalpy obtained as a function of the re-heating temperature with annealing T at 45 °C for different times. This annealing temperature was within the vicinity of DSC T_g_, which was relatively close to the DSC T_f_, thus indicating that the relaxation should be considerably faster at this T than at 8 °C. According to Götze MCT, T_a_ was lower than T_g_; thus, the α relaxation mode has been stretched out (slowed down), and the τ_2_ relaxation mode cannot be neglected [25]. With the system approaching T_g_ (45 °C), these multiple relaxation times, including β, α, and beyond, were shortened, and the mode slowdown effect was minimized. This starting annealing point was similar to the “c” point on the blue dashed line exhibited in Figure 2a, and the relaxation process moved toward the equilibrium line, which was closer to the quench line. Therefore, T_p_ shifted to a low temperature, and relaxation enthalpy decreased with the increase in annealing time. In particular, the spacing between different relaxation times might be scaled by a single temperature-dependent coefficient α_T_, and all relaxation times would closely follow that in the Arrhenius region. This procedure would provide modes a chance to relax to their equilibrium state of 45 °C. After annealing for 5 min, the long-wavelength modes relaxed to a state similar to the regenerated film due to chain stretching. For the results of the regenerated film displayed in Figure 3, the annealing curves for 2 min were already similar to the melt curve, but the curve of the electrospun membrane under the same condition still deviated considerably from the melt curve. The relaxation process for the regenerated film finished considerably faster than the electrospun membrane from the DSC observation. With the increase of aging time, the enthalpy decreases and T_p_ shift to low temperature tend to be different between the fiber membrane and the regenerated membrane. The enthalpy decrease in fiber membranes was similar to the regenerated film, but T_p_ was nearly constant at the concise aging times (less than 2 min). This trend was due to the lack of long relaxation modes constrained by stretched polymer chains and the lack of a large surface area in regenerated films, requiring energy for melting (the large surface per mass) similar to the electrospun membrane. The phenomena due to the interplay of slow (long wavelength) and fast (short wavelength) relaxation modes at temperatures slightly above or in the vicinity of DSC T_g_ could not be predicted by any equilibrium glass transition theories nor by the truncated Götze MCT. Glass transition was controlled entirely by relaxation dynamics or non-equilibrium thermodynamics. Nevertheless, Götze MCT paved the way for realizing the actual physical picture and control parameters in applications.

A previous study on the relaxation process has focused on a temperature range far lower than T_g_ in Region I of Figure 2b to observe the slowdown of long wavelength modes [26]. To compare the distinctions caused by slight annealing temperature differences within the vicinity of T_g_ in Region II, the PLGA-regenerated film and fiber films were all annealed for 1 min at various temperatures, and the results were presented in Figure 3. The DSC curves of different annealing temperatures were consistent with the relaxation processes’ thermodynamics evidence. The variations in the relaxation processes were due to the transformation from non-equilibrium to an equilibrium state, typically entailed an increase in temperature or lengthening the time. The variation trend of the T_p_ and the relaxation enthalpy depicted in Figure 3 were consistent with the results demonstrated in Figure 1 and Figure 2.

As previously discussed, the annealing temperature was within the vicinity of the DSC T_g_ or above this T_g_; thus, the slower relaxation modes stretched out initially, and the slow, long-wavelength relaxation modes could not be neglected. The long-wavelength relaxation modes could be gradually relaxed toward their equilibrium states with the increase in the annealing temperature. The potential energy variation determined relaxation enthalpy during the intersegmental distance fluctuation, related to the chain mobility variation. However, this distance was initially shortened for electrospinning confinement during residual solvent evaporation and was lengthened afterward to increase the repulsive force during the relaxation process and inter-molecular packing fluctuation related to the stretched long-chain relaxation. When the inter-segmental distance was shortened, the repulsive force must be increased, causing the long-chain relaxation to lengthen further. Therefore, the relaxation enthalpy of the electrospun membranes was initially increased then decreased by raising the annealing temperature with a fixed 1 min annealing time. However, the relaxation enthalpy of the regenerated film was directly decreased by increasing the annealing temperature because no limit was observed given the stretched long-chain relaxation. This rare relaxation process of the electrospun membrane could be observed and analyzed considering the fiber’s chain mobility limitation and confined long-wavelength relaxation modes. Therefore, relaxation enthalpy and T_p_ change can also be observed easily. Relaxation enthalpy and T_p_ initially increased then decreased with the annealing temperature in Region II (Figure 2b). This electrospun membrane had been demonstrated as a suitable model system for studying the dynamics of the glass transition process, which qualitatively follows the direction of the MCT (not truncated).

### 3.4. Physical Aging of the PLGA Electrospun Membrane with Different Molecular Weights around Region II

The comparative experiments between the regenerated film and electrospun membranes illustrated that the relaxation process would move toward the equilibrium line, regardless of the aging temperature. All samples were stored at 8 °C for a long time before the DSC aging experiments. The starting point was on the Aging-1 Line below the “a” point (as represented by the blue dashed line in Figure 2a). If T_a_ was lower than T_g_ in Region I (from the “a” point to the equilibrium line), then the enthalpy loss or free volume decrease would be increased with aging time but would proceed to the opposite, whereas T_a_ was around T_g_ in Region II (from the “b” point where the equilibrium line was above the aging line in this region). Although the shifting of T_p_ results showed no significant difference with the decrease or increase in the enthalpy loss within several aging measurements, FVT or truncated MCT could explain the results if ascribed to the sensitivity of the DSC instrument. To understand the thermodynamic relaxprocesses quantitatively, three samples with molecular weights from 20 k to 60 k were selected for aging experiments at the DSC glass transition region. When a high annealing temperature in Region III (Figure 2b) was used, even a short experimental annealing time (even 1 min) was extensive relative to all the relaxation times of the small molecular weight samples; therefore, the structure evolution by the discontinuous temporal evolution of DSC curves was difficult to capture. However, a long annealing time (even a couple of hours) was too short relative to the relaxation process of the same sample annealed in Region-I because it can hardly produce any observable relaxation evolution. To directly address the measurement limitation of instrument sensitivity and timescale, the most sensible option for annealing temperatures was in Region II. The appropriate instantaneous states during the relaxation process could be obtained by adequately selecting polymer molecular weights to understand and predict the actual relaxation process. The three PLGA molecular weights were larger than the molecular entanglement weight and could ensure a sufficient intermolecular interaction to generate the membrane with uniform fibers of similar fiber diameters. Here, the molecular entanglement weight for different molecular weights of the PLGA membrane (in the solid-state) was assumed to be the same; moreover, the spinning solutions were all highly concentrated, and their viscosity was similar to obtaining the same fiber diameter. Furthermore, all the PLGA samples in this study were synthesized under the same condition with the same monomer ratio.

If the relaxation process was independent of the long-wavelength modes that originated from the stretched long-chain polymers, then the trends of the T_p_ and relaxation enthalpy of different M_w_ samples should be similar to each other at the same aging conditions. However, as illustrated in Figure 4a–c, the change in T_p_ and the relaxation enthalpy for the low molecular weight membrane was fast under the same annealing temperature (42.5 °C, as a typical example). The DSC curve of M_w_ = 20 k membrane (Figure 4a), which was annealed for 2 min, was nearly the same as the melt curve. However, the M_w_ 100,000 g/mol membrane (Figure 4c) remained deviated considerably from the melt curve even after annealing for 100 min. Two factors, namely, the annealing temperature difference from the measured DSC T_g_ and the number of confined long-wavelength modes (i.e., the entanglement density and the molecular weights in the electrospun fibers), determined the molecular weight dependency. The DSC T_g_ was typically defined as the beginning temperature during the temperature increase scan, in which the endothermic enthalpy starts to deviate from the baseline. This deviation could be interpreted as the fast or short-wavelength modes that start to follow the temperature scanning instrument, absorb heat, and relax toward their liquid/melt states. However, the long-wavelength modes had not relaxed at this temperature. Therefore, the DSC T_g_ and enthalpy loss were related to the influence of short-wavelength modes (or fast relaxation times) on these membranes because the thermal difference obtained through the DSC was mainly determined by the change in the intermolecular potential energy. The heating curves’ peak temperatures were the measurements of long wavelengths or slow relaxation modes of these membranes because long-wavelength modes cannot catch up with the increase in temperature during the re-heating process at a low temperature. The long-wavelength modes in the electrospun membrane were further extended by the long-stretched chain conformation during the fiber spinning, although the intermolecular interaction should still be propagated (or correlated) from local to long distances. The variation in the DSC T_g_ reflected the beginning of the severe melting of relaxation modes, and the DSC T_p_ marked the beginning of the end of slow relaxation modes (long wavelength melting) from different molecular weights of polymers in the sample. The fibers with low molecular weight easily became relaxed with a slight enthalpy loss and T_p_ at the aging temperatures given the small entanglement density. However, the T_p_ of the fibers with higher molecular weight increased, whereas the enthalpy loss decreased. Therefore, the short-wavelength modes have been relaxed at this aging temperature, whereas the long-wavelength modes still existed and were confined by the long-distance stretched polymer chains, thus resulting in a delayed T_p_ during the re-heating process.

After annealing at various temperatures (in Region II of Figure 2b) for 1 min, the temperature evolutions of the relaxation process of PLGA fibers with different molecular weights were measured and were displayed in Figure 4d–f. Such a wide annealing temperature range could provide an improved perspective and a possible prediction of a real dynamic relaxation process. When the annealing temperature was discreetly increased by 1 min in Region II of Figure 2b, the fast relaxation modes were initially relaxed, and the long-wavelength relaxation modes gradually emerged and relaxed afterward. This phenomenon was evident when the electrospun membrane was used as the annealing material (Figure 4d–f). The curves gradually shifted to the melt curve by raising the annealing temperature (in T_a_ > T_g_, T_a_ was 37.5–42.5 °C in Figure 4d, 42.5–47.5 °C in Figure 4e, and 47.5 °C in Figure 4f) for the different molecular weights of the membrane and was caused by the gradual relaxation of long-wavelength modes. However, the relaxation enthalpies were all initially increased, and T_p_ remained constant when the annealing temperature was raised (in T_a_ < T_g_, T_a_ was 35 °C in Figure 4d and 40 °C in Figure 4e), and decreased afterward with PLGA of a relatively low molecular weight (20 and 60 k). Moreover, the trends of relaxation enthalpy and T_p_ of 100 k PLGA were displayed differently at this annealing temperature region (Figure 4f). When the short wavelength modes have been relaxed, but the long-wavelength modes were confined with the stretched polymer chain at a long distance, the results were similar with PLGA (20 and 60 k) fibers at low aging temperature (37.5 °C in Figure 4f), but the T_p_ increased with enthalpy loss decreased at a relatively high aging temperature (42.5 °C and 45 °C in Figure 4f).

### 3.5. Data Analysis and General Discussion of the Relaxation Process of PLGA Fibers with Different Molecular Weights

Following Figure 2b, physical aging experiments were performed in Region II at various annealing temperatures using different molecular weight membranes as materials, and the temperature and temporal evolution were analyzed in detail. The aging temperatures for each molecular weight were individually set in their region around the DSC T_g_. Expressly, for M_w_ 20 k materials, aging temperatures were set from 30–47.5 °C, and for M_w_ 60 k and 100 k materials, aging temperatures were set from 37.5–55 °C. The aging temperatures gaps were 2.5 °C. The aging times were set to 0.5, 1, 2, 5, 10, and 100 min. In Figure 5, the T_p_ and enthalpy of each re-heating curve after annealing were collected to determine the regularity of the relaxation processes of PLGA fibers with different molecular weights. Although all the fibrous membranes underwent the same aging process before the DSC annealing experiments, the fibers with lower molecular weight initially presented a high enthalpy loss and a low T_p_ (the spots at 0 min in Figure 5, without DSC annealing) given its relatively low DSC T_g_, minimal entanglement densities, and small confinements in the stretched polymer chain. This result indicated that the short wavelength relaxation modes, mostly relaxed or close to their equilibrium states during the initial sample storage at 8 °C, were dominant. In addition, the fluctuations in the intermolecular distance shifted to a close distance. After annealing at a certain temperature for a specific time in the DSC chamber, the temporal evolutions of the T_p_ value were displayed in Figure 5(a1–a3). T_p_ values were increased by increasing the aging time at a low annealing temperature and were consistent with the previous conclusion exhibited in Figure 1. The T_p_ values were initially increased and then directly decreased with the continuous annealing time and temperature increase. This result was also consistent with the conclusion displayed in Figure 4. Given the increase in the annealing temperature, additional long-wavelength relaxation modes, which were frozen initially (due to a slowed down relaxation time), showed a relaxation effect, and the intermolecular distances were initially decreased and then lengthened by following the fluctuation of the intermolecular potential curve. Specifically, the T_p_ values of the re-heating curves of 20 k PLGA remained constant when T_a_ < T_g_ (30–35 °C in Figure 5(a1)) and decreased substantially when T_a_–T_g_ (37.5–42.5 °C in Figure 5(a1)) with the increase in annealing time. However, with the increase in annealing time to more than 10 min at an even low temperature (35 °C), the T_p_ and the corresponding enthalpy loss began to decrease (Figure 5(b1)). The enthalpy loss decreased rapidly when T_a_ > T_g_ increased annealing time, indicating that the short wavelength modes without sufficient confinement were sensitive to high temperatures in the PLGA fibers with low molecular weight. The values of T_p_ and enthalpy loss in 60 k PLGA increased when T_a_ < T_g_ (37.5 °C and 40 °C in Figure 5(a2,b2)), and then the enthalpy loss began to decrease when T_a_–T_g_ (42.5–47.5 °C in Figure 5(b2)) with an increase in annealing time. However, the T_p_ value in this aging region increased and subsequently decreased with an increase in annealing time. The values of T_p_ and enthalpy loss in 100 k PLGA samples slightly increased when T_a_ < T_g_ (37.5–42.5 °C in Figure 5(a3,b3)), thus denoting that the long-wavelength modes at a long distance cannot be relaxed at this temperature region and limit the relaxation of short-wavelength modes even with an annealing time of more than 100 min. With an increase in annealing temperature when T_a_–T_g_ (45 °C in Figure 5(a3,b3)), T_p_ and enthalpy loss initially increased and then directly decreased with the increase in annealing time. This phenomenon might be due to the relaxation of long-wavelength modes, causing the loss of confinement on short wavelength and resulting in the subsequent decrease. When T_a_ > T_g_, regardless of the size of the molecular weight, the T_p_ and enthalpy loss of the PLGA fibers would directly decrease, and the curves would shift toward the melted one.

For the same fiber diameter, the confined number of long-wavelength relaxation modes increased by increasing the molecular weight, short relaxation times in the spectrum distribution for a low molecular weight sample, and long-wavelength slow modes in the spectrum distribution large molecular weight sample. Therefore, the fluctuation in the relaxation properties was mainly determined by the region where annealing was performed (Figure 2b). Within region I, the short-wavelength modes dominated the relaxation process, which resulted in the variation in enthalpy loss being more evident than T_p,_ and both of the enthalpy loss and T_p_ would be increased with annealing time (limited to month time). Within Region II, the relaxation modes with different wavelengths created a competitive and compliance relationship, and the variation in T_p_ and enthalpy loss showed noticeable inflections with the increase in annealing time. With the increase in annealing temperature, the effects of long-wavelength relaxation modes, which were initially negligible, had become evident, and the confinement would gradually be released during the relaxation process. The fluctuation rule at this region was entirely different from the previous ones, even for a short annealing time, in which the variation in T_p_ would display the different trends from enthalpy loss. The variation in enthalpy loss and T_p_ that relied on the molecular weight and chain entanglement density demonstrated the existence and importance of multiple relaxation modes during the aging process of glassy polymer materials. Furthermore, the guidance for applications can only be obtained from a qualitative analysis of relaxation dynamics until a quantitative theoretical solution for a condensed system was discovered.

The exact relationship of the inflection point with different molecular weights was displayed in Figure 6a, in which samples with the annealing time of 10 min were selected. The annealing temperature was used as a horizontal axis to demonstrate the change in inflection points directly, as presented in Figure 6. This inflection point was probably the annealing temperature at that annealing time when most short wavelength modes have already relaxed to their instantaneous equilibrium states, and the remaining long-wavelength modes require a long time to relax or a high temperature to accelerate the relaxation. The inflection point illustrated in Figure 6a shifted to a high annealing temperature by increasing the M_w_ from 20 k to 100 k. The annealing temperature of the inflection point was high for the increased molecular weight. If the polymers consist of the exact proportions of co-monomers, samples of different molecular weights have the same entanglement molecular weight (Me). Therefore, the relaxation spectral distribution was due to chain size, their response to the spinning process, and changes in entanglement density at different molecular weights [30,31]. Therefore, the number of entanglement points per chain was small for the low molecular weight sample, or the entanglement density of low molecular weight polymer was low. Thus, the chain mobility decreased for a high entanglement density and required additional energy to finish the orientation relaxation for high molecular weight samples. Therefore, the inflection point shifted to a high temperature for the annealing temperature, and T_p_ was illustrated in Figure 6a.

In terms of the DSC-defined T_g_, the DSC curves were divided into three regions for convenience: Region I (far below T_g_), Region II (annealing temperature close but around T_g_, can be below or above T_g_), and Region III (annealing temperature far above T_g_) in Figure 2b. Only five relaxation times were depicted in Figure 6c to illustrate the mode slowdown principle of a dynamic glassy system with the decrease in the environment (or laboratory) temperature. If the temperature keeps in Region II (e.g., T_3_ in Figure 6c), several long-wavelength modes may relax slower than the instrument time, which will lag and cause an apparent structure solidification in a reasonable laboratory timescale, thereby resulting in the aging phenomenon. If the temperature remains in Region-I (much lower than T_5_), certain long-wavelength modes will have extensive relaxation times, and the macroscopic shape or specific volume of the materials will have no apparent change with a long time or will change slowly with time.

Since all samples were subjected to a constant time low-temperature annealing treatment, a new equilibrium has been reached for fast relaxation modes involving short wavelengths or other local motion, but no significant relaxation for slow relaxations involving long wavelengths. Suppose this experiment was performed through the regenerated film. In that case, only one macroscopic deformation mechanism can work for polymer chains in the regenerated film given the decrease in temperature and solvent evaporation, thereby decreasing the thickness and causing the compression in the z-direction of polymer chains. Their fast modes were easily relaxed or orientated but cannot be identified with the DSC enthalpy peak from the aging process or treatment (as illustrated in Figure 6d, lower plot). Only the α and β modes can be observed, depending on the measuring instrument’s temperature or time scale sensitivity. However, for the fibrous structure, two major mechanisms, namely, the spinning process and solvent evaporation, can produce chain deformation in forming the membrane [26]. In addition to the local stretch, the polymer chain was highly stretched under a long length scale (such as an end-to-end distance or a radius of gyration) and a high density, which requires additional energy to relax (or melt) to the corresponding bulk state (as demonstrated in Figure 6d, upper plot). During the relaxation process, additional stretched local structures will be compressed and propagated, thus affecting the non-relaxed long-wavelength modes and causing a further slowdown of the long-wavelength or slow modes until the long-wavelength modes were exposed to high temperature or annealing for an extended time can be relaxed. Considering that a new starting point for a new temperature scanning path was the instantaneous state of the sample, the shape of the new path was the sum of the total enthalpy absorbed by all the dynamic modes.

## 4. Summary and Conclusions

The PLGA electrospun fibrous membranes with three molecular weights and unique fiber diameters were used as a model system with regenerated films of the same composition. Without the availability of a quantitative MCT, structural relaxation processes were analyzed semi-quantitatively. In this study of a confined relaxation system, a normal DSC instrument with a typical temperature scanning rate of 20 °C/min which was sufficient for investigating the dynamic relaxation process through an adequately designed kinetic scanning procedure. The proper use of DSC measurements obtained the “arrested” instantaneous structures with different annealing temperatures and times through a gradual propagation of the relaxation processes of the electrospun membranes and their regenerated films. Short wavelength modes contributed to the potential energy, which can be measured using the ΔH of DSC curves, whereas long-wavelength modes mainly contributed to T_p_. Therefore, the constrained and stretched PLGA polymer chains provided additional long-wavelength relaxation modes and clear and extended temperature range and timescale for glass transition and aging studies through DSC experiments. The aging process could be divided into three regions, and the special features in each region were concluded in Table 2.

This study investigated the dynamic relaxation behavior of glass transition, physical aging, and shape memory effect. This study demonstrated that glass transition was a dynamic process controlled by a multimode relaxation process. A complete MCT-type theory was necessary to analyze the time-nonequilibrium aging process of polymeric glassy materials. The detailed investigations on the relaxation processes of PLGA fibers with three molecular weights also provided a quantitative understanding of the space-time correlation in the glass transition process and the control parameters to tailor the shape memory function release in special applications, such as biomedical devices.

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
