# Peer review of "Electrospun Fibrous Membrane with Confined Chain Configuration: Dynamic Relaxation and Glass Transition"

_polymers, 2022, doi:10.3390/polym14050939_

Round 1
Reviewer 1 Report
polymers-1584887
This paper reports development of Electrospun Fibrous Membrane with Confined Chain Configuration by focussing on Dynamic Relaxation and Glass Transformation. The paper is fundamentally strong and brings important information on the handling of nanofiber. The paper is technically sound with only minor flaws in the writing. Few comments below can be addressed to enhance the manuscript quality.
- Abstract need an introduction and some quantitative data.
- There are still many typos in the manuscript. Many headings’ titles need to be corrected.
- The research gap is not clearly presented. Justify clear;y the novelty of the work.
- Figure 1d: The measured fiber diameter seems to be much smaller than the one shown in the SEM image.
- Many of the measurements seem to be done once. How to justify the statistical significance of the results?
- Figs 5 are too small to read.
- Section 4 does not provide any conclusion. Please rewrite.
- The descriptive model in Table 2 is very interesting. However, justification from other works is required to support the proposal.
Author Response
To Reviewer 1
Comment1. Abstract need an introduction and some quantitative data.
Answer: Thanks for your comment. The abstract of manuscript has been rewritten to show the novelty and the relevance of the work. An introduction and some quantitative data also added into the abstract.
Comment2.There are still many typos in the manuscript. Many headings’ titles need to be corrected.
Answer: Thank you for your suggestion. As suggested by reviewer, the language presentation was improved with assistance from a native English speaker with appropriate research background.
Comment3. The research gap is not clearly presented. Justify clearly the novelty of the work.
Answer: Thanks for your comment. The abstract of manuscript has been rewritten to show the novelty and the relevance of the work.
Comment4. Figure 1d: The measured fiber diameter seems to be much smaller than the one shown in the SEM image.
Answer: We deeply appreciate the reviewer’s suggestion. According to the reviewer’s comment, the SEM image in Figure 1 was reconfirmed. The mean fiber diameter ± SD values of fiber membranes of different molecular weights are added in Figure 1.
Comment5. Many of the measurements seem to be done once. How to justify the statistical significance of the results?
Answer: Thanks for your comment. All experiments were repeated. The data are statistically significant. The authors modified some of the graphs in Figure 1 to reflect this.
Comment6. Figs 5 are too small to read.
Answer: Thanks for your comment. Figure 5 has been re-edited for clearer reading
Comment7. Section 4 does not provide any conclusion. Please rewrite.
Answer: Thanks for your comment. The conclusion section has been re-edited.
Comment8.The descriptive model in Table 2 is very interesting. However, justification from other works is required to support the proposal.
Answer: Thanks for your comment. This section has been rewritten and re-edited into the Conclusions section. Previous work supporting this conclusion has been added to the manuscript

Reviewer 2 Report
Title: Electrospun Fibrous Membrane with Confined Chain Configuration: Dynamic Relaxation and Glass Transformation
Zhang et al. studied the enthalpy relaxation of spun PLGA fibers.
Although the authors performed some tests and presented corresponding results, there are still some issues that need to be addressed before further consideration.
- The title is generic. The Authors should consider adding the studied system
- The Authors should consider modifying unfitting terms such as “dynamic relaxation” and “glass transformation” present in the title, abstract and text
- The introduction does not give an adequate background. Also, enthalpy relaxation is a property of PLGA itself, as film and spheres, see Yoshioka 2011, Allison 2007, Bailey 2002, and Rouse 2007. Reference 14 does not match the discussion.
- Line 59-60: this aspect in not properly discussed in the results and discussion section
- Line 30-32: the polyurethane used in the references are thermosetting, which does not fit the discussion.
- Line 44-54: it is not clear how the MCT theory has been used in this work
- Spinning concentrations are not reported
- Line 116-118: PLGA is an amorphous polymer. WAXS measurements are not reported
- Line 125: The fibres reported are in the micrometers scale.
- Line 165: Reference 28 does not match the topic
- It is not clear what the Author refer to while speaking about short- and long-wavelength modes
- Figure 4: the caption is misleading. 1a-3a should be replaced with Figure 4a-4c, 1b-3b should be replaced with Figure 4d-4f
- Line 419: Figure 5 (1a-3a) should be replaced with Figure 5 (a1-a3), same for line 443.
- Line 503: typing error
- The conclusions are not supported by the findings
Author Response
To Reviewer 2
Comment1. The title is generic. The Authors should consider adding the studied system
Answer: Thanks for your comment. But Electrospinning systems were already mentioned in the title.
Comment2. The Authors should consider modifying unfitting terms such as “dynamic relaxation” and “glass transformation” present in the title, abstract and text
Answer: Thank you for your suggestion. As suggested by reviewer, the language presentation was improved with assistance from a native English speaker with appropriate research background.
Comment3. The introduction does not give an adequate background. Also, enthalpy relaxation is a property of PLGA itself, as film and spheres, see Yoshioka 2011, Allison 2007, Bailey 2002, and Rouse 2007. Reference 14 does not match the discussion.
Answer: Thanks for your comment. PLGA is a popular biodegradable material1 and is a US Food and Drug Administration-approved polymer. Over the past two decades, PLGA has been a rapidly increasing research interest given its intrinsic biodegradability and bioassimilability2. A PLGA electrospun fibrous membrane normally has a fiber diameter in the nanometer scale and a high surface area/volume ratio with enhanced properties. The polymer chains are physically constrained and confined in the electrospun fibers of the membrane to a certain scale. Thus, the polymer chains are not only important in a membrane application but also a favorable model for studying the glass transformation and aging processes in general.
Inappropriate expressions have been corrected. Part of the above theory was added to the manuscript to make the theory appear more complete. References have been revised, and references mentioned by reviewers have been cited as appropriate.
- Xu, Y.; Kim, C. S.; Saylor, D. M.; Koo, D., Polymer degradation and drug delivery in PLGA-based drug-polymer applications: A review of experiments and theories. Biomed. Mater. Res. B. Appl. Biomater. 2017, 105 (6), 1-25.
- Thomas, C. M.; Lutz, J. F., Precision synthesis of biodegradable polymers. Angew Chem. Int. Ed. Engl. 2011, 50 (40), 9244-9246.
Comment4. Line 59-60: this aspect in not properly discussed in the results and discussion section
Answer: Thank you for your suggestion. The PLGA-regenerated film does not have a porous structure, so this part of the conclusion is obvious. The authors supplement the description of regenerated PLGA membranes in this section to make the theory clearer
Comment5. Line 30-32: the polyurethane used in the references are thermosetting, which does not fit the discussion.
Answer: Thanks for your comment. But the example of the aging of a polyurethane coating was to illustrate physical aging of glassy polymer materials, which is below their transition temperature leads to a risk. Even though the polyurethane used in the references are thermosetting, the aging process is still carried out below the transition temperature.The citation is not inappropriate.
Comment6. Line 44-54: it is not clear how the MCT theory has been used in this work. Spinning concentrations are not reported
Answer: Thank you for your suggestion.
As an important but vague issue1–2, the glass transition of structural materials have been studied and discussed well in four most popular theories3–6: (1) free volume theory (FVT), (2) entropy theory, (3) mode-coupling theory (MCT), and (4) caging and jamming theory (CJT). Vogel–Fulcher–Tammann (VFT) theory7-8 and William–Landau–Ferry (WLF) theory9 are two major variations of FVT, and the two theories are phenomenological theories that work well above the conventionally defined (i.e., differential scanning calorimetry [DSC]-measured) glass transition temperature Tg, in which the viscosity is scaled with a single-temperature-dependent relaxation time. Although WLF theory considers multiple relaxation times (a relaxation spectrum), all relaxation times are assumed to scale with a single-temperature-dependent function αT in terms of temperature. In particular, all relaxation times maintain their proportionality with each other throughout all the temperatures. Therefore, the dynamic (or modes) slowing down of glassy materials, and consequently the physical aging, cannot be properly accounted for in these theories. Entropy theory10-11 is an equilibrium theory linked to dynamics through Adams and Gibbs theory12 (or ansatz), which has been successful in equilibrium or near-equilibrium cases; however, problems on extrapolating to glass transition temperature and beyond exist. Furthermore, including the dynamics and mode slowdown process are infeasible. CJT may be closely linked to experimental observations and simulations/computations of colloidal systems and is closely related to the results of Götze MCT13. Formal MCT starts from a Zwanzig–Mori projection operator expansion of a generalized dynamic equation14. N relaxation times constantly appear for an N-particle system, but Götze MCT has truncated all high-order correlation terms after the second term (fourth-order correlation in the reciprocal space) because of a complicated operation. Consequently, Götze MCT15 has retained only a β-mode and an α-relaxation branch mode. The truncation of MCT typically misses certain important physics, especially the slow relaxation times, which are important in the slowing-down and glass-aging processes at temperatures below the conventional DSC Tg and at a timescale slower than a regular instrumental measuring time.
Experimentally, the relaxation process through the glass transition region is normally investigated with stress and dielectric relaxation16, scattering13 and differential volumetric techniques17, and DSC18. A colloid system is typically investigated by dielectric relaxation, scattering techniques, and sometimes, high-resolution optical microscopy. The entanglement and polymer conformational relaxation processes can be observed through stress relaxation. The DSC approach can show the thermal history for the physical aging process and the relaxation evolution by tracing the results of the endothermic temperature process and relaxation enthalpy, respectively. The calorimetry or DSC curves has been proven useful in providing information from the beginning of the heating or cooling process because these curves establish a link to the free energy change history of the glassy sample. Hopefully, through various temperature scanning and annealing of our fibrous membrane with a confined, long-chain configuration, the temporal and structural changes or the enthalpy changes or relaxations can be studied and understood.
Inappropriate expressions have been corrected. Part of the above theory was added to the manuscript to make the theory appear more complete.
- Yuan, G.; Cheng, H.; Han, C. C., The glass formation of a repulsive system with also a short range attractive potential: A re-interpretation of the free volume theory. Polymer 2017, 131 (22), 272-286.
- Hodge, l. M., Physical aging in polymer glasses. Science 1995, 267 ( 5206), 1945-1947.
- Eckert, T.; Bartsch, E., Re-entrant glass transition in a colloid-polymer mixture with depletion attractions. Rev. Lett. 2002, 89 (12), 125701-125704.
- Pham, K. N.; Puertas, A. M.; Bergenholtz, J.; Egelhaaf, S. U.; Moussaid, A.; Pusey, P. N.; Schofield, A. B.; Cates, M. E.; Fuchs, M.; Poon, W. C., Multiple glassy states in a simple model system. Science 2002, 296 (5565), 104-106.
- Foffi, G.; De Michele, C.; Sciortino, F.; Tartaglia, P., Scaling of dynamics with the range of interaction in short-range attractive colloids. Rev. Lett. 2005, 94 (7), 078301-078304.
- Yavari, M.; Maruf, S.; Ding, Y.; Lin, H., Physical aging of glassy perfluoropolymers in thin film composite membranes. Part II. Glass transition temperature and the free volume model. Membrane Sci. 2017, 525, 399-408.
- Fulcher, G. S., Analysis of recent measurments of the viscosity of glasses. Am. Ceram. Soc. 1925, 8, 339-355.
- Schonhals, A.; Goering, H.; Schick, C.; Frick, B.; Zorn, R., Glassy dynamics of polymers confined to nanoporous glasses revealed by relaxational and scattering experiments. Phys. J. E Soft Matter 2003, 12 (1), 173-178.
- Malcolm L. Williams; Robert F. Landel; Ferry, J. D., The temperature dependence of relaxation mechanisms in amorphous polymers and other glass-forming liquids. Am. Chem. Soc. 1955, 77, 3701-3707.
- Gibbs, J. H.; Dimarzio, E. A., Nature of the glass transition and the glassy State. Chem. Phys. 1958, 28, 373-384.
- Dudowicz, J.; Freed, K. F.; Douglas, J. F., Generalized entropy theory of polymer glass-formation. Chem. Phys. 2008, 137, 125-224.
- Adam, G.; Gibbs, J. H., On the Temperature Dependence of Cooperative Relaxation Properties in Glass‐Forming Liquids. The Journal of Chemical Physics 1965, 43 (1), 139-146.
- Glotzer, S. C., Spatially heterogeneous dynamics in liquids insights from simulation. Non-Cryst. Solids 2000, 274, 342-355.
- Van Hove, L., Correlations in space and time and born approximation scattering in systems of interacting particles. Rev. 1954, 95 (1), 249-253.
- Kob, W., The Mode-Coupling theory of the glass transition/supercooled liquids. ACS: 1997; Vol. 676, p 28.
- Kaminski, K.; Kipnusu, W. K.; Adrjanowicz, K.; Mapesa, E. U.; Iacob, C.; Jasiurkowska, M.; Wlodarczyk, P.; Grzybowska, K.; Paluch, M.; Kremer, F., Comparative study on the molecular dynamics of a series of polypropylene glycols. Macromolecules 2013, 46 (5), 1973-1980.
- Villiger, M.; Lorenser, D.; McLaughlin, R. A.; Quirk, B. C.; Kirk, R. W.; Bouma, B. E.; Sampson, D. D., Deep tissue volume imaging of birefringence through fibre-optic needle probes for the delineation of breast tumour. Rep. 2016, 6, 28771-28781.
- Andreozzi, L.; Faetti, M.; Giordano, M., Molecular-weight dependence of enthalpy relaxation of PMMA. Macromolecues 2005, 38, 6056-6067.
Comment7. Line 116-118: PLGA is an amorphous polymer. WAXS measurements are not reported
Answer: Thanks for your comment. But This manuscript investigates the relationship between material molecular weight and relaxation mode and the materials used have similar crystallinity. And the crystallinity of the material is not a variable to be investigated in this manuscript. It can be seen from the SEM images that the microstructure types of the materials are highly similar.
Comment8. Line 125: The fibres reported are in the micrometers scale.
Answer: Thank you for your suggestion. Inappropriate expressions have been corrected.
Comment9. Line 165: Reference 28 does not match the topic
It is not clear what the Author refer to while speaking about short- and long-wavelength modes
Answer: Thanks for your comment. From the dynamic perspective, the motions of all molecules and/or segments are coupled in condensed systems. If a complete dynamic equation can be written down and solved for an N-particle system1,2, 3, then an N-relaxation time (or modes) theoretical model can be developed. However, solving an N-particle condensed system problem is nearly infeasible at present. Although Götze MCT 2 is a truncated theory, it only provides two modes (α and β) for glass transformation and demonstrates an accelerated slowdown of the α branch (or mode) relative to the β relaxation time in a simple system. However, the importance of the dynamic modes and the slowdown of long wavelength modes3 are the most important characteristics of any glassy materials in a temperature below the DSC-measured Tg and in the release of the shape memory function in applications for a glassy material. Traditionally, the dynamics of the glass transformation of a polymer material is considered a process from a thermodynamically equilibrium liquid state to a frozen-in thermodynamically nonequilibrium glass state4–6. When the liquid is cooled through the glass transition region, molecular mobility slows down, and the material may become a slow-varying but unstable glass.
Inappropriate expressions have been corrected. Part of the above theory was added to the manuscript to make the theory appear more complete.
- Yuan, G.; Cheng, H.; Han, C. C., The glass formation of a repulsive system with also a short range attractive potential: A re-interpretation of the free volume theory. Polymer 2017, 131 (22), 272-286.
- Kob, W., The Mode-Coupling theory of the glass transition/supercooled liquids. ACS: 1997; Vol. 676, p 28.
- Capaccioli, S.; Thayyil, M. S.; Ngai, K. L., Critical issues of current research on the dynamics leading to glass transition. Phys. Chem. B 2008, 112, 16035-16049.
- Solar, M.; Binder, K.; Paul, W., Relaxation processes and glass transition of confined polymer melts: A molecular dynamics simulation of 1,4-polybutadiene between graphite walls. Chem. Phys. 2017, 146 (20), 203308-203322.
- Cangialosi, D.; Alegría, A.; Colmenero, J., Effect of nanostructure on the thermal glass transition and physical aging in polymer materials. Polym. Sci. 2016, 54-55, 128-147.
- Cohen, L., Caregiving stress, endogenous sex steroid Hormone levels, and breast cancer incidence. Soft Matter 2009, 20 (1), 919-926.
Comment10. Figure 4: the caption is misleading. 1a-3a should be replaced with Figure 4a-4c, 1b-3b should be replaced with Figure 4d-4f
Answer: Thank you for your suggestion. Inappropriate expressions have been corrected.
Comment11. Line 419: Figure 5 (1a-3a) should be replaced with Figure 5 (a1-a3), same for line 443.
Answer: Thank you for your suggestion. Inappropriate expressions have been corrected.
Comment12. Line 503: typing error
Answer: Thanks for your comment. The language presentation was improved with assistance from a native English speaker with appropriate research background.
Comment13. The conclusions are not supported by the findings
Answer: Thanks for your comment. The conclusion section has been re-edited.Some parts of section 3 has been rewritten and re-edited into the Conclusions section. Previous work supporting this conclusion has been added to the manuscript

Reviewer 3 Report
Peer-Review polymers – 1584887
The manuscript entitled “Electrospun Fibrous Membrane with Confined Chain Configuration: Dynamic Relaxation and Glass Transformation” by Nuozi Zhang et al. constitutes a well-design study of the dynamics relaxation during glass transition comparing PLGA electrospun fibrous membranes with PLGA-regenerated films. The manuscript is well written, properly organized and fits within the scope of the journal Polymers (ISSN 2073-4360), particularly in the special issue “Advances of Polymeric Membranes”. However, few issues must be addressed before its consideration for publication.
Major issues:
- The authors should identify clearly the novelty and the relevance of the manuscript in the abstract or in the final part of the Introduction section.
- (Materials and Methods, subsection 2.2.): In the description of the electrospinning process, the authors refer that they produced the PLGA electrospun fibers using a broad range of values for the voltage (16-24 kV), flow rate (10-20 μL/min) and tip to collector distance (18-28 cm). Since the variation of these parameters has a high influence on the fiber structure and diameter, it is better to report the specific parameter values used to fabricate the PLGA membranes.
- In Figure 1, the authors should add the values for average fiber diameter ± SD for the different molecular weight fibrous membranes. Moreover, contact angle results should be mentioned in the Figure 1 caption. The number of fibers measured in the analysis should also be reported.
Minor issues:
- Please use capital letters in the manuscript main sections: 1. Introduction (line 25), 2. Experimental section (line 65), 3. Results and Discussion (line 122), 4. Summary and Conclusions.
- (Page 2, line 73): It is better to add the word “system” to obtain: “(...) a mixed solvent system of acetone and DMF (...)”.
- (Page 7, Figure 2 caption, line 220): Please remove “:”.
- (Page 8, line 272): It is better to replace the “;” by “, and”.
- (Page 15, line 543): Please add the Celsius symbol: 20°C/min.
Author Response
To Reviewer 3
Major issues:
Comment1. The authors should identify clearly the novelty and the relevance of the manuscript in the abstract or in the final part of the Introduction section.
Answer: Thanks for your comment. The abstract of manuscript has been rewritten to show the novelty and the relevance of the work. An introduction and some quantitative data also added into the abstract.
Comment2. (Materials and Methods, subsection 2.2.): In the description of the electrospinning process, the authors refer that they produced the PLGA electrospun fibers using a broad range of values for the voltage (16-24 kV), flow rate (10-20 μL/min) and tip to collector distance (18-28 cm). Since the variation of these parameters has a high influence on the fiber structure and diameter, it is better to report the specific parameter values used to fabricate the PLGA membranes.
Answer: Thanks for your comment. More detailed process parameters have been added in the manuscript to better demonstrate the preparation of electrospun fiber membranes.
Comment3. In Figure 1, the authors should add the values for average fiber diameter ± SD for the different molecular weight fibrous membranes. Moreover, contact angle results should be mentioned in the Figure 1 caption. The number of fibers measured in the analysis should also be reported.
Answer: We deeply appreciate the reviewer’s suggestion. According to the reviewer’s comment, a description of the detailed fiber diameter and water contact angle data has been added in Section 2.3.
Minor issues:
Comment1. Please use capital letters in the manuscript main sections:
(1) Introduction (line 25),
(2) Experimental section (line 65),
(3) Results and Discussion (line 122),
(4) Summary and Conclusions.
Answer: We deeply appreciate the reviewer’s suggestion. According to the reviewer’s comment, the language presentation was improved with assistance from a native English speaker with appropriate research background.
Comment2. (Page 2, line 73): It is better to add the word “system” to obtain: “(...) a mixed solvent system of acetone and DMF (...)”.
Answer: We deeply appreciate the reviewer’s suggestion. According to the reviewer’s comment, the language presentation was improved with assistance from a native English speaker with appropriate research background.
Comment3. (Page 7, Figure 2 caption, line 220): Please remove “:”.
Answer: We deeply appreciate the reviewer’s suggestion. According to the reviewer’s comment, the language presentation was improved with assistance from a native English speaker with appropriate research background.
Comment4. (Page 8, line 272): It is better to replace the “;” by “, and”.
Answer: We deeply appreciate the reviewer’s suggestion. According to the reviewer’s comment, the language presentation was improved with assistance from a native English speaker with appropriate research background.
Comment5. (Page 15, line 543): Please add the Celsius symbol: 20°C/min.
Answer: We deeply appreciate the reviewer’s suggestion. According to the reviewer’s comment, the language presentation was improved with assistance from a native English speaker with appropriate research background.

Round 2
Reviewer 2 Report
The comments have been addressed